# Possible Relationship between Long-Term Adverse Health Effects of Gonad-Removing Surgical Sterilization and Luteinizing Hormone in Dogs

**DOI:** 10.3390/ani10040599

**Published:** 2020-04-01

**Authors:** Michelle A. Kutzler

**Affiliations:** Department of Animal and Rangeland Sciences, Oregon State University, 112 Withycombe Hall, Corvallis, OR 97370, USA; michelle.kutzler@oregonstate.edu

**Keywords:** behavior, canine, luteinizing hormone, musculoskeletal, neoplasia, neuter, obesity, spay, urinary incontinence

## Abstract

**Simple Summary:**

Spaying and neutering dogs is commonly used to prevent the birth of unwanted animals. However, spaying and neutering is associated with an increased risk of several long-term health problems including obesity, urinary incontinence, bladder stones, hypothyroidism, diabetes mellitus, hip dysplasia, cruciate ligament rupture, behavioral changes (including owner-directed aggression and fear), cognition problems, as well as several forms of cancer (including leukemia, prostate cancer, bone cancer, skin cancer, splenic cancer, and bladder cancer). An explanation of how spaying and neutering increases the risk of these long-term health problems is discussed in this review.

**Abstract:**

Spaying and neutering dogs is commonly used to prevent the birth of unwanted animals and eliminate the risk of reproductive diseases. However, removal of the gonads prevents the feedback of estrogen and testosterone on the pituitary and hypothalamus. As a result, luteinizing hormone (LH) is continuously elevated at supraphysiologic concentrations. Although the main role of LH is for reproductive function (e.g., ovulation), there are LH receptors present in several normal tissues including the thyroid and adrenal glands, gastrointestinal tract, cranial cruciate ligament and round ligament, and lymphocytes. In addition, there are LH receptors present in several neoplastic tissues (e.g., lymphoma, hemangiosarcoma, mastocytoma, transitional cell carcinoma, and osteosarcoma). The role of LH receptors in non-reproductive normal and neoplastic tissues is not known but may stimulate nitric oxide release and induce cell division. The precise etiology of the increased incidence of several non-reproductive long-term health complications following spaying and neutering is not known but may be related to LH receptor activation in these non-reproductive target tissues. How these effects may be mediated is described in this review.

## 1. Introduction

Throughout most of the developed world, surgical sterilization has become a common tool for combatting the overpopulation of unwanted dogs and eliminating the risk of reproductive diseases in pet dogs (e.g., mammary gland cancer and prostate hyperplasia/infection) [1,2,3,4,5,6]. In the United States, 64% of dogs have been surgically sterilized, and this is most commonly performed between 6 weeks and 12 months of age [7]. For this review, ovariectomy and ovariohysterectomy (spay) or castration (neuter) will be collectively referred to as gonadectomy, since each of these methods for surgical sterilization includes gonad removal (ovaries or testes).

In the normal adult mammal, the hypothalamus secretes gonadotropin-releasing hormone (GnRH), which stimulates the anterior pituitary gland to release luteinizing hormone [8]. Luteinizing hormone (LH) stimulates the secretion of gonadal steroid hormones (testosterone in males and estrogen/progesterone in females). These gonadal steroid hormones then negatively feedback to the hypothalamus and anterior pituitary to decrease the secretion of GnRH and LH, respectively. However, in the gonadectomized mammal, there is no negative feedback, which results in supraphysiologic circulating concentrations of LH. In gonadectomized dogs, LH concentrations are more than thirty times the concentrations found in normal adult dogs [9]. Although the main role of LH is for reproductive functions (e.g., ovulation, corpus luteum formation), there are LH receptors present throughout the body, not just limited to the reproductive tract. The purpose of LH receptors in non-reproductive tissues is not known but may induce cell division and stimulate nitric oxide release [10]. With constant activation following gonad removal, these receptors can be upregulated (unpublished observations), further magnifying the effects of the supraphysiologic LH concentrations in non-reproductive tissues. The following review summarizes several non-reproductive long-term health complications resulting from spaying and neutering as well as discusses the possibility of how these effects are mediated by LH receptor activation in these non-reproductive target tissues.

## 2. Non-Neoplastic Disorders

Obesity is a serious medical problem defined as an excessive accumulation of fat beyond the physical and skeletal limits [11]. Gonad removal is the single largest risk factor for the development of obesity in dogs [12]. Up to 68% of spayed and neutered dogs are obese [13,14,15,16,17,18]. Gonadectomy induces obesity through two main mechanisms: increased appetite and decreased metabolic rate. Gonad removal stimulates food intake [19] and increases indiscriminate appetite [20]. In unaltered dogs, food intake suppresses the secretion of gastrointestinal hormones (cholecystokinin and glucagon), resulting in satiety (alleviation of hunger) [21]. However, within 1 week following de-sexing, food intake increases by 20% and then persists [22,23]. It is possible that stimulation of LH receptors (present in the gastrointestinal tract following gonadectomy) increases cholecystokinin and/or glucagon release. It is also possible that LH receptors in the hypothalamus are involved in the increase in appetite [24], as lesions within the ventromedial hypothalamus result in hyperphagia [25].

Urinary incontinence is an involuntary leakage of urine resulting from either a weakened or complete loss of urinary sphincter control. The association between urinary incontinence and gonad removal in female dogs was first described by Jo (1965) [26]. Urinary incontinence is a common long-term health complication of spaying female dogs, with a reported incidence ranging from 5% to 30% [27,28,29,30]. Early-age spaying (under 5 months of age) may further increase the risk of occurrence of urinary incontinence [30,31] but the association between the age at spaying and the development of incontinence is weak. LH receptors are expressed in all regions of the canine lower urinary tract, from the body and neck of the bladder to the proximal and distal urethra [32,33]. Spayed female dogs with urinary incontinence have a significantly higher number of LH receptors in the lower urinary tract compared with unaltered females [34]. Urinary continence can be restored in spayed females using estrogens [29,35,36,37,38], GnRH agonists [39,40], or GnRH immunization [41,42]. These treatments all decrease circulating LH concentrations.

Urinary calculi are solid particles (concretions) in the urinary system, usually composed of mineral salts that can form in any part of the urinary tract [43]. After evaluating records from more than two million dogs, Banfield Pet Hospital found that all urinary calculi (urine crystals, kidney stones, and bladder stones) occurred at a rate three times higher in spayed and neutered dogs compared with unaltered dogs [44]. Under normal circumstances, there is a balance of urinary calculi promoters and inhibitors. However, this balance appears to be disrupted from the influence of an abundant LH environment.

Diabetes mellitus results from the impaired secretion of insulin with variable degrees of peripheral insulin resistance leading to hyperglycemia. In dogs, the incidence of diabetes mellitus is 0.4%–1.2% [45] and has been increasing over the past 30 years [46,47]. Gonad removal doubles the risk for developing diabetes mellitus in dogs [46]. Although gonadectomy increases the risk for obesity, the increased prevalence for diabetes mellitus in spayed and neutered dogs is unrelated to obesity [48,49] and may be a direct effect of LH on the pancreas (e.g., chronic pancreatitis).

Hypothyroidism is a common endocrine disorder in which the thyroid gland does not produce sufficient quantities of thyroid hormone [50,51]. Gonad removal has a profound effect on thyroid function [52] and is the most significant cause for the development of hypothyroidism in dogs [53]. Thirty percent more spayed and neutered dogs develop hypothyroidism compared with unaltered dogs [54]. The concentrations of thyroxine in spayed and neutered dogs were significantly lower in both genders when compared with intact dogs [55]. Women who have undergone gonadectomy are also at an increased risk for developing hypothyroidism [56]. LH receptors are expressed in normal and adenomatous human thyroid glands [57]. Our laboratory has reported on the presence of LH receptors in the canine thyroid gland co-localized with thyroid stimulating hormone (TSH) receptors [58]. It is possible that continuous LH receptor activation is interfering with the mechanism of action of TSH in the thyroid, resulting in hypothyroidism.

Canine hip dysplasia is associated with the abnormal joint structure and laxity of the muscles, connective tissue, and ligaments that would normally support the hip [59,60,61]. As hip joint laxity increases, the articular surfaces between the acetabulum and the head of the femur lose contact with each other, resulting in subluxation. Over time, subluxation results in a significant change in the size and shape of both articular surfaces and varying severity of osteoarthritis. It is important to note that most dogs with hip dysplasia are born with normal hips but then develop hip dysplasia secondary to intrinsic and/or extrinsic factors. The incidence of hip dysplasia can be as high as 40%–83% in giant and large breed dogs [59,62,63]. However, the incidence of hip dysplasia varies considerably between different giant and large dog breeds. Independent of the occurrence of obesity, gonadectomy significantly increases the incidence of canine hip dysplasia [64]. Compared to unaltered dogs, gonadectomy increases this by 1.5 times [59] to 2 times [65] the occurrence in unaltered dogs. It is important to mention that the positive relationship between gonadectomy and the incidence of hip dysplasia is strongest in neutered male dogs as opposed to spayed female dogs. The mechanism for the increased incidence in certain breeds or sexes is not known but our laboratory has demonstrated the expression of LH receptors within the ligament of the head of the femur, the hyaline cartilage, and subchondral bone of the femur head [66]. It is possible that, in some dogs, an increase in LH receptor activation in the structural support tissues within the hip joint results in increased laxity, which is responsible for the higher occurrence of hip dysplasia in spayed and neutered dogs. However, our preliminary evidence has not shown a significant difference in LHR expression between sexes or reproductive statuses in these structural support tissues.

The cranial cruciate ligament serves to prevent cranial displacement of the tibia relative to the femur, to limit internal rotation of the tibia relative to the femur, and to prevent stifle hyperextension [67,68]. Cranial cruciate ligament rupture is another musculoskeletal disorder that initially involves the degeneration of the cranial cruciate ligament, which leads to a partial rupture and then progresses to a complete rupture following an unspectacular traumatic event [69,70]. Similar to hip dysplasia, most dogs with cranial cruciate ligament ruptures are born with normal stifle joints but then develop the tendency for cranial cruciate ligament rupture secondary to intrinsic and/or extrinsic factors. Gonad removal significantly increases the prevalence of cranial cruciate ligament rupture [71], doubling the occurrence reported for unaltered dogs [72], with an incidence as high as 5.1% and 7.7% in males and females, respectively [65]. Prepubertal spaying and neutering delays tibial growth plate closure, which extends the length of tibia and the steepness of the tibial plateau [73,74]. Increased steepness of the tibial plateau can increase the cranial tibial thrust, which is a risk for cranial cruciate ligament rupture [75,76]. Despite the skeletal deformations that occur with prepubertal gonadectomy (under 6 months), even dogs that are gonadectomized between 6 and 12 months have an increased risk for cranial cruciate ligament rupture [65]. There is some evidence that hormones (estrogen and relaxin) may play a role in altering cranial cruciate ligament laxity and modify risk factors in humans [77,78]. Our laboratory has demonstrated the expression of LH receptors within the cranial cruciate ligament [66]. It is possible that an increase in LH receptor activation in the cranial cruciate ligament results in increased laxity, which is responsible for the higher occurrence of ligament ruptures in spayed and neutered dogs.

The role of gonad removal on behavior is complex. Evidence for benefits as well as detriments following gonadectomy has been reported. Reproductive-related behaviors (such as urine marking in house, mounting, and roaming) are all reduced or eliminated following gonadectomy [79,80,81]. However, fear and aggression tend to be exacerbated [82]. Fear of storms, fear of gunfire, fear of noises, fear biting, timidity, separation anxiety, and submissive urination all increase significantly following spaying or neutering. Spayed females are also more reactive to the presence of unfamiliar humans and dogs [83]. Although some dogs may become less aggressive following gonadectomy [80], dominance aggression [84] and owner-directed aggression [20,85] occur with a significantly higher frequency in gonadectomized dogs compared with unaltered dogs. The hippocampus and hypothalamus both play important roles in controlling behaviors, especially those pertaining to fear and aggression. Luteinizing hormone receptors are abundant in the hippocampus and hypothalamus [86,87,88]. In addition, administration of supraphysiologic concentrations of LH to gonadectomized animals can induce aggression and other behavioral changes [89,90,91].

Cognitive dysfunction syndrome is a neurodegenerative disorder of senior dogs, which is characterized by both cognitive changes and neurophysiological pathologies [92,93]. Memory impairment, poor problem-solving skills, social disconnect, confusion, and day–night reversal may occur as the condition progresses. Gonad removal significantly increases the development and progression of cognitive dysfunction syndrome in dogs [94]. Increases in luteinizing hormone are associated with declines in cognitive performance [95]. In addition, elevated LH concentrations increase beta amyloid plaque formation and are implicated in the development of Alzheimer’s syndrome in humans [96,97]. Therefore, it is possible that LH and its receptor are important in the development of cognitive dysfunction syndrome in spayed and neutered dogs.

## 3. Neoplastic Disorders

Unlike the condition in men, the aggressive nature of the canine prostate adenocarcinoma and the lack of a screening test make the identification of early-stage prostate cancer in dogs extremely problematic [98]. In dogs, gonadectomy is the largest risk factor for the development of prostate adenocarcinoma [99,100]. Luteinizing hormone receptors are abundant in the prostate gland and increase in expression following gonadectomy [101,102,103]. Prostate carcinomas in dogs are associated with a high rate of metastasis at presentation and poor prognosis even with aggressive local therapies [98]. Prostatectomy is associated with significant postoperative morbidity without significantly extending survival times [104,105].

Transitional cell carcinomas can arise from the bladder or urethra, including the prostatic urethra [106,107,108,109,110]. Even with surgical removal, radiation treatment and chemotherapy, the prognosis for dogs with transitional cell carcinomas is poor, with only 16% of treated dogs surviving for over one year [111]. Gonadectomized dogs have a significantly higher risk of developing a transitional cell carcinoma compared with unaltered dogs [111]. Luteinizing hormone receptors are widely distributed throughout the bladder and urethra and increase in expression following gonadectomy [32,33,112,113]. Our laboratory has also demonstrated the abundant expression of LH receptors in transitional cell carcinoma tissue.

Osteosarcoma is a highly metastatic cancer of bone tissue. Despite many advances over the past 20 years, survival times for dogs diagnosed with osteosarcoma have not changed, with the principal cause of mortality being the development of pulmonary metastases [114]. Osteosarcoma occurs with significantly higher frequency in spayed and neutered dogs [115]. The incidence of osteosarcoma in gonadectomized Rottweiler dogs is 1.3–2.0 times higher than in unaltered dogs [116]. However, there were no differences in the incidence of osteosarcoma between gonadectomized and unaltered German Shepherds. It is not known whether LH receptors exist in the bone or whether LH could be using an indirect mechanism to mediate the increased incidence of osteosarcoma.

Hemangiosarcoma is a rapidly growing, highly invasive cancer arising from the lining of blood vessels and occurring almost exclusively in dogs. Primary tumors can arise in any vascular tissue, but the spleen and heart are the most common locations for hemangiosarcoma to develop. Even with surgical removal, the mean life expectancy is 86 days (range, 10–202 days) without adjunctive chemotherapy and 189 days (range, 118–241 days) with adjuvant chemotherapy [117]. Many studies have confirmed the presence of LH receptors in vascular endothelial and smooth muscle cells [118,119]. Spayed female dogs have two times the risk for developing splenic hemangiosarcoma and five times the risk for developing cardiac hemangiosarcoma compared with unaltered females [82,120]. Our laboratory has also demonstrated the expression of LH receptors in hemangiosarcomas, which may explain why this cancer is more common in spayed females [121].

Mastocytoma is the most common skin tumor in dogs [122]. Luteinizing hormone receptors are abundant in the skin [32,123]. Several studies have documented an increased risk for developing mastocytoma following spaying or neutering in dogs [65,82,124]. Our laboratory has shown that not only do mastocytomas express LH receptors, but that these tumors express three distinct patterns of LH receptor immunoexpression [125]. Moreover, mastocytomas from gonadectomized dogs had significantly higher more LHR-positive mast cells (84.2 ± 8.7%) overall. In addition, LHR-positive mast cells exhibiting the type 2 pattern (66.6 ± 15.3%) compared with mastocytomas from intact dogs (64.3 ± 4.2% and 49.2 ± 8.4%, respectively) [125]. The higher expression of LHR provides a mechanism that could be exploited in intervention strategies (e.g., using GnRH agonists) for mastocytoma recurrence in spayed and neutered dogs, leading to prolonged survival time.

Lymphoma is a cancer of lymphocytes and/or lymphoid tissues. Lymphoma is the most common cancer diagnosed in dogs, accounting for up to 24% of all canine cancers [126]. LH receptors are present in lymphocytes and lymphoid tissue (medulla of thymus) [127,128]. Our laboratory has demonstrated that the mean percentage of circulating LH receptor-positive T lymphocytes is significantly higher in gonadectomized dogs (16.6%) than in sexually intact dogs (10.5%); whereas the percentages of circulating LH receptor-positive B lymphocytes did not significantly differ by reproductive status [128]. Gonadectomy increases the incidence of lymphoma [82]. In Golden Retrievers, neutered males are three times more likely to develop lymphoma than unaltered males and approximately 1 in 10 neutered males will develop lymphoma [65]. Our laboratory has demonstrated that 12.4% of cells in canine neoplastic lymph nodes expressed LH receptors [128]. In addition, we showed that in vitro activation of LH receptors on T-lymphoma cells stimulates cell proliferation [129].

## 4. Conclusions

This review has focused the long-term adverse health effects of gonad-removing surgical sterilization in dogs in general. This review has not included differences in the occurrence of these adverse effects between age at spaying/neutering (early versus late), breed differences, or between sex differences. There is limited research addressing these points and a need for future research in these areas, especially as this research relates to differences in LHR expression. It is also important to note that, at present, there is no direct proof that there is a causative connection between elevated LH concentration/LHR expression and the long-term adverse health effects of spaying/neutering discussed in this review. More research on the causative connection between elevated LH concentration/LHR expression and the long-term adverse health effects of spaying/neutering is needed.

In addition, this review was focused on the long-term adverse health effects of gonad-removing surgical sterilization specifically in dogs. Although cats also suffer from long-term adverse effects of elevated LH concentrations following spaying/neutering (e.g., polyphagia, aggression and anxiety, diabetes), at the current time, the benefits of gonad removal (e.g., elimination of urine marking, recurrent/persistent estrus, mammary adenocarcinoma) outweigh these detriments. 

Furthermore, this review has focused on the relationship between the long-term adverse health effects of gonad removal and elevated LH concentration/LHR expression. This review has not mentioned what, if any, effect that elevated follicle-stimulating hormone (FSH) concentrations have on the long-term adverse health effects of gonad removal. Although FSH concentrations are equally high in spayed/neutered dogs, FSH has extragonadal negative and positive feedback via pituitary inhibin and activin, respectively, whereas LH concentration has no extragonadal feedback. Additional research is needed on what role FSH concentrations/FSH receptors may play in the long-term adverse health effects of spaying/neutering.

Unrelated to any particular disease or major cause of death, years of gonad retention prolong longevity in a population of Rottweiler dogs [130]. Based upon the review of the literature, it becomes clear that canine gonads are not merely reproductive organs but are critical to endocrine, musculoskeletal, behavior, and anti-neoplastic health. Among the non-reproductive functions of gonads, suppression of LH secretion and resulting LH receptor overexpression appear necessary in maintaining homeostasis. Therefore, a surgical sterilization method that enables the dog to keep gonads intact while still preventing reproduction (e.g., ovary-sparing hysterectomy, vasectomy) may prolong their health, especially since the risk of fatal disease associated with retaining the gonads (mammary cancer, prostatic enlargement) is relatively low. In addition, research investigating the effect of LH/FSH down-regulation using a long-acting GnRH agonist (e.g., deslorelin) may provide a treatment option to mitigate or prevent the long-term adverse health effects of spaying/neutering.

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
