# Peer review of "Possible Relationship between Long-Term Adverse Health Effects of Gonad-Removing Surgical Sterilization and Luteinizing Hormone in Dogs"

_animals, 2020, doi:10.3390/ani10040599_

Round 1

Reviewer 1 Report

The review has been adapted according to my suggestions and is now acceptable for publication

Author Response

Thank you for your assistance in reviewing this manuscript.

Reviewer 2 Report

Animals  Manuscript ID  animals-706158-R1

Title: Systematic review of the long-term adverse health effects of gonad removing surgical sterilization in dogs

In general the manuscript improved, all concerns were addressed. I suggest publication provided the minor points below are corrected.

Line 52: remove the word are

Line 79: Again I’d be carefull with the wording since estrogen probably in addition affects via its receptor and not solely via LH. A suggestion: “Urinary incontinence can be restored in spayed females by using estrogens, GNRH agonists or GNRH immunization. These medicaments have in common that they decrease the LH concentration”.

Author Response

The two changes recommended by the reviewer were made.

This manuscript is a resubmission of an earlier submission. The following is a list of the peer review reports and author responses from that submission.

Round 1

Reviewer 1 Report

This is a very thorough review of the research literature related to spay/neuter. The main contribution is to explore, to the extent possible, receptor sites of LH in non-reproductive tissue, presumably linking the possibility of some abnormal occurrences of organ systems, joint disorders, and cancers to abnormal activation of receptor sites is a major contributio

The author does not seem to acknowledge the effect of long-term loss of gonadal hormones. Also, the age of spay/neuter, which relates to the length of time the dog has been exposed to high levels of LH, was not mentioned. Dogs of some breeds seem not to be affected by spay/neuter with regard to some cancers or joint disorders or mammary cancers. The author's work on LH makes one wonder if the increased risk of cancers and joint disorders seen in dogs of some breeds is a reflection of high LH levels, low levels of gonadal hormones, or both.

The author is an expert on LH levels and receptor sites. What would be useful would be to look at different situations – early spay/neuter vs. later spay/neuter; breed differences; sex differences; etc. There is little research addressing these points but they could be outlined as needing future research if LH elevation is to be considered part of the issue in addressing the issues around spay/neuter. As long as dogs are around, we will be dealing with spay/neuter. And, of course, one then wonders about cats; although, for example, males will always be neutered, are they  living with high LH levels?

Author Response

The reviewer brings up a good point about not acknowledging the cursory effects of long-term loss of gonadal hormones. However, the rise in LH hormone that is the focus on this manuscript is the main (and in my opinion, most important) effect of the long-term loss of gonadal hormones.

The reviewer also makes a good point related to the age of spay/neuter.  Most epidemiologic studies that report an increased incidence in long-term health problems following gonad removal do not include the age of spay/neuter so it is not possible to answer this question. However, the difference in exposure to elevated LH concentrations if spayed/neutered under 6 months of age or under 12 months of age is irrelevant (+/- 6 months) given the remaining years of lifetime exposure to LH.

The reviewer mentioned that there may be a breed effect of LHR in the different tissues that may be responsible for the increased risk of cancers and joint disorders seen in dogs of some breeds. I certainly agree that this is an area needed for future research.

Although cats suffer from elevated LH concentrations (e.g. polyphagia, aggression and anxiety, diabetes, etc.), at the current time, the detriments of gonad retention outweigh the detriments of gonad removal.

Each of these points has been addressed in the conclusion of the revised manuscript.

Reviewer 2 Report

General comments

This is a very timely review paper, showing important data on the occurrence and possible mechanisms for increased orthopedic problems and cancers in neutered and spayed dogs, from a specialist in the field. However, I have a few general comments which need to be included before the paper is acceptable for publication:

Although there are indications from the literature and from the author’s lab that increased LH-receptor expression is present in several organs/cancers of spayed dogs as compared to control intact animals, there is at present no proof that there is a causative connection. This should be clearly mentioned in the conclusions followed by a call for more research on this topic. Moreover, there is no mentioning of FSH-levels, which are equally high in neutered dogs. Are they not important? What about AMH levels which are low in these dogs? Such novel research could also include to investigate the effect of a deslorelin implant, a long acting GnRH-agonist (which may not be available in the USA?), that is reducing LH levels, on cancers in neutered dogs. I suggest to rephrase the conclusion which is now : “Therefore, a surgical sterilization method that enables the dog to keep gonads intact while still preventing reproduction is likely to prolong their health.” This is too strongly expressed. Veterinarians will oppose against such an approach (vasectomy, salpingectomy) for fear of pyometra, mammary tumours and prostatic disease. It may be wise to add at some point in the review the incidence and prevalence of these diseases, so it is clear that the risk is relatively low, and to mention that these diseases are less fatal than the cancers induced by spaying and neutering. It would be useful to add a figure with the possible mechanisms of continuous high LH-levels on different canine organs and cancer development.

Specific comments

Line 115 : please add punctuation point/full stop at the end of the sentence

Line 128 - Remove the 1 behind closure: Prepubertal spaying and neutering delays tibial growth plate closure,1 which extends the length of tibia and the steepness of the tibial plateau

Author Response

The reviewer makes a good point that at present there is no proof that there is a causative connection between elevated LH concentration/LHR expression and the long-term adverse health effects of spay/neuter. More research in this area is needed.

The reviewer also makes a good point that there is no mentioning of FSH-levels, which are equally high in neutered dogs. Unlike FSH which has extragonadal negative and positive feedback via pituitary inhibin and activin, respectively; LH has no extragonadal feedback. However, additional research is needed on the role FSH concentrations /FSH receptors may play in the long-term adverse health effects of spay/neuter.

The conclusion statement was revised as suggested by the reviewer.

Line 115 : A punctuation point has been added at the end of the sentence

Line 128 : The "1" in this sentence was deleted.

Reviewer 3 Report

This review provides an overview of the literature about a very hot topic in small animal veterinary medicine. I completely agree that many of the described side effect of castration might be partly or completely related to LH-secretion but more research is needed to confirm this for many of these conditions. Some nice hypothesis are presented here in this review about the link between LH and the described condition. However in this review I miss a nuanced representation of the literature published about this topic so far. For example, some of the articles cited focused their research on a specific breed, whereas in this review their conclusions are presented as a conclusion for the whole dog population. Besides that, it has been shown already that the age of neutering has a major impact on the prevalence of some of these conditions, however this is not mentioned in this review.

Additional remarks:

Title: I think the term 'systematic review' is not a correct definition for this literature overview. I would propose "Potential relationship of high LH -levels with post-neutering disease in dogs"

Line 25: there is a 'bracket' missing after osteosarcoma

Line 37: What about the common age of spaying in the US? Is it typically done in very young dogs?

Line 52: is there a reference article for the this finding (up-regulation of the LH-receptors?)

Line 73: link of the age of castration seems debatable according to the systematic review of Beauvais et al., 2012

Line 86: Other hypothesis could be: chronic pancreatitis after surgical intervention?

Line 111: in reference 64: the impact of castration on hip dysplasia was only found for male dogs, not for females. / You mention the studies on boxers and golden retrievers, but not the one on labradors and German Shepherds. / Is there an impact of the age of castration?

Line 131: in the reference mentioned (65): an increased odd is described if the dog is neutered at the age of < 12 months, but not in late neutering. I'm a bit struggling here with the terminology 'post pubertal', to what extent we can call this post-pubertal? // what about the studies on labrador and german shepherd?

Line 180: the study mentioned here is on rottweilers, however for German Shepherds, no differences in the prevalence of osteosarcoma were found. 

Line 212: The study mentioned is about golden retrievers and not 'the dog' in general. Maybe it's worth while mentioning this.

Author Response

The conclusion has been revised as suggested by the reviewer.

The title has been revised.

Line 25: The 'bracket' was closed in this sentence.

Line 37: An age range was provided as requested.

Line 52: Unpublished (ongoing) observation added to this sentence.

Line 73: This sentence has been revised.

Line 86: This sentence has been revised.

Line 111: This sentence has been revised.

Line 131: This sentence has been revised.

Line 180: This sentence has been revised.

Line 212: This sentence has been revised.

Reviewer 4 Report

In this review, the author summarizes health problems probably associated with gonadectomy, at least occurring more frequently in gonadectomized than in intact dogs. An increased expression of LH receptors in non-reproductive tissues of gonadectomized dogs is inferred to contribute to some of the diseases. Findings from studies with canine tissues are compared with findings from human medicine. The author hypothesizes that sterilization leaving the gonads intact might prevent some of the non-reproductive long-term health complications described.

In general

The theme is highly relevant, and the previous findings and hypotheses of the author very interesting, however, will require  further investigations. Nevertheless, from my point of view the review is worth publishing, provided the concerns mentioned below are considered.

The Introduction resembles too much the abstract, may be more data from the controversial literature around the discussion “to castrate or not castrate” should be provided here – and the probable impact of prepubertal castration.  Maybe also the combined effects of certain factors like body weight and time of castration – very brief. The fact that in castrated dogs the LH concentrations are significantly higher and possible consequences may then be highlighted and lead over to the review (line 40…)

Line 57: Obesity is a serious

Line 91: is there any hint to proof this interesting hypothesis – some human studies or with mice?

Line 101: Could it be just the decrease in estrogens? I would just mention some suggestions and theories from other authors and then mentions your theory. The list of facts is interesting but I miss more info from the cited studies.

Line 115: it is not obvious whether in the cited study gonadectomized dogs or intact dogs were examined. Do you know whether the LH receptors are upregulated in these tissues after castration?

Line 179 and 190: is this independent from the time of gonadectomy? Similar in prepubertally and postpubertally gonadectomized dogs?

Line 217: …prolonged longevity in a population of Rottweiler dogs.

Line 222: I would choose a more careful wording like…is likely to prevent some of the diseases that mainly occur in gonadectomized dogs (think about mammary tumors and pyometra in intact dogs)

Author Response

The Introduction was lightly revised but given the space limitation for this manuscript, there is not much opportunity to make substantial changes to these two paragraphs.

Line 57: This sentence was revised as suggested by reviewer.

Line 91: I am not aware of any research in other species to support this.

Line 101: Other studies have introduced theories (e.g. lack of E2) but there is no evidence to support this.  For this reason, the only theories that are introduced in this review are those with at least some evidence to support them.

Line 115: This sentence was revised.

Line 179 and 190: This sentence was revised.

Line 217: This sentence was revised.

Line 222: The conclusion was revised.